

# Residual uncertainty estimation using instance-based learning with applications to hydrologic forecasting

Omar Wani[1,2,a], Joost V.L. Beckers[2], Albrecht H. Weerts[2,3], Dimitri P. Solomatine[1,4]

[1]UNESCO-IHE Institute for Water Education, Delft, the Netherlands
[2]Deltares, Delft, the Netherlands
[3]Hydrology and Quantitative Water Management Group, Department of Environmental Sciences, Wageningen University, Wageningen, the Netherlands
[4]Water Resource Section, Delft University of Technology, Delft, the Netherlands
[a]currently at: Institute of Environmental Engineering, Swiss Federal Institute of Technology (ETH), Zurich and Swiss Federal Institute of Aquatic Science and Technology (Eawag), Dübendorf, Switzerland

*Correspondence to*: O. Wani (owani@student.ethz.ch)

## Abstract

A non-parametric method is applied to quantify residual uncertainty in hydrologic streamflow forecasting. This method acts as a post-processor on deterministic model forecasts and generates a residual uncertainty distribution. Based on instance-based learning, it uses a k-nearest neighbour search for similar historical hydrometeorological conditions to determine uncertainty intervals from a set of historical errors, i.e. discrepancies between past forecast and observation. The performance of this method is assessed using test cases of hydrologic forecasting in two UK rivers: Severn and Brue. Forecasts in retrospect were made and their uncertainties were estimated using kNN resampling and two alternative uncertainty estimators: Quantile Regression (QR) and Uncertainty Estimation based on local Errors and Clustering (UNEEC). Results show that kNN uncertainty estimation produces accurate and narrow uncertainty intervals with good probability coverage. The accuracy and reliability of these uncertainty intervals are at least comparable to those produced by QR and UNEEC. It is concluded that kNN uncertainty estimation is an interesting alternative to QR and UNEEC for estimating forecast uncertainty. An advantage of this method is that its concept is simple and well understood, it is relatively easy to implement and it requires little tuning.

## 1 Introduction

Hydrologic forecasts for real-life systems are inevitably uncertain (Beven and Binley, 1992; Gupta et al., 1998; Refsgaard et al., 2007). This is due to the uncertainties in the meteorological forcing, in the modelling of the hydrologic system response and in the initial state of the system at the time of forecast. It is well accepted that, compared to a simple deterministic forecast, additional information about the expected degree of accuracy of that forecast is valuable and generally leads to better decision making (Krzysztofowicz, 2001). Various techniques have therefore been developed to quantify uncertainties





associated with the meteorological model input (van Andel et al., 2013), the initial state of the model (Li et al., 2009) and with the hydrologic models themselves (Deletic et al., 2012; Coccia and Todini, 2011). Frameworks and guidelines have been developed to incorporate uncertainty analysis of environmental models effectively in decision making (Arnal et al., 2016; Reichert et al., 2007; Refsgaard et al., 2007). Broadly, there are three basic approaches to uncertainty estimation: i)

explicitly defining a probability model for the system response e.g. (Todini, 2008), ii) estimation of statistical properties of the error time series in the post-processing phase of model forecast e.g. (Dogulu et al., 2015) and iii) methods using Monte Carlo sampling of inputs and/or parameters, aimed at quantifying the output probabilistically e.g. (Beven and Binley, 1992; Freer et al., 1996). Many uncertainty estimations techniques employ a combination of these approaches (Montanari and Brath, 2004; Del Giudice et al., 2013). Some techniques focus on one source of uncertainty, such as the model parameter

uncertainty (Benke et al., 2008) or the model structure uncertainty (Butts et al., 2004), while others focus on combined uncertainties stemming from model parameters, model structure deficits and inputs (Schoups and Vrugt, 2010; Evin et al., 2013; Del Giudice et al., 2013). In this context, it is important to note that apart from estimating uncertainty of model parameters during calibration, uncertainty estimation for hydrologic forecasting requires quantification of predictive uncertainty, which includes uncertain system response to new combinations of model input parameters (Renard et al., 2010;

Coccia and Todini, 2011; Dotto et al., 2012).

In this paper, we will restrict ourselves to the class of uncertainty estimators  called post-processors. These methods usually do not discriminate between different sources of uncertainty. They "aggregate" all sources into a so-called residual uncertainty. Post-processing methods assume the existence of a single calibrated model with an optimal set of model

parameters, and build a statistical or machine learning model of the residual uncertainty. Typically, these techniques relate a combination of model inputs and/or outputs to the model error distribution. Various post processors have been developed and applied to hydrologic modelling, such as Quantile Regression (Weerts et al., 2011), UNEEC (Solomatine and Shrestha, 2009) and DUMBRAE (Pianosi and Raso, 2012). Quantile Regression (QR) is a relatively straightforward post-processing technique that relates the probability of residual errors to the model forecast (the predictand) by a regression model that is

derived from historical forecasts and observations. QR has been successfully applied for uncertainty quantification in hydrologic forecasts with various modifications (Weerts et al., 2011; Verkade et al., 2013; Roscoe et al., 2012; López López et al., 2014; Hoss and Fischbeck, 2015). Whereas, UNEEC involves a machine learning technique for building a non-linear regression model of error quantiles (Solomatine and Shrestha, 2009). UNEEC includes three steps: 1) Fuzzy clustering of input data in the space of "relevant" variables; 2) Estimating the probability distribution function of residual errors for each

cluster and 3) building a machine learning model (e.g. an artificial neural network) of the prediction interval for a given probability (Dogulu et al., 2015). Many other uncertainty estimation techniques, such as DUMBRAE (Pianosi and Raso, 2012), HUP (Krzysztofowicz, 1999), Model Conditional Processor (Coccia and Todini, 2011), Bayesian revision (Reggiani et al., 2009) and Bayesian Model Averaging (Raftery et al., 2005) make assumptions about the nature of the probability distribution function of error. This is not necessary for QR and UNEEC (Lopez et al., 2014; Dogulu et al., 2015).



Nevertheless, in QR and UNEEC assumptions need to be made about the form of the regression function that is used to calculate the quantiles.

In an attempt to explore the utility of easier-to-implement post-processing techniques, we employ a simple non-parametric forecast method for residual uncertainty quantification. This method uses kNN search to learn about the past residual errors, which avoids having to make explicit assumptions about the nature of the error distribution and tuning of distribution parameters. Instance-based learning has been used in meteorology and hydrology before for resampling of precipitation and streamflows, most notably by Lall and Sharma (1996), who used the k-nearest neighbour (kNN) method for resampling of monthly streamflow sequences. kNN search has also been used in a non-parametric simulation method to generate random sequences of daily weather variables (Rajagopalan and Lall, 1999). They defined a weighting function for probability where the predictand is resampled from k values. Jules and Buishand (2003) used nearest-neighbour resampling to generate multi-site sequences of daily precipitation and temperature in the Rhine basin. Also, instance-based learning has been used as a data-driven model for hydrologic forecasting (Solomatine et al., 2008;Solomatine and Ostfeld, 2008). Beckers et al. (2016) use nearest neighbour resampling to generate monthly sequences of climate indices and related precipitation and temperature series for the Columbia River basin. Specifically in the context of error modelling, a version of UNEEC that uses kNN instance-based learning as its basic machine learning technique to predict the residual error quantiles, was compared to the original ANN-based UNEEC in Shrestha and Solomatine (2008). However, kNN can be also used without the complicated UNEEC procedure that includes fuzzy clustering. The application of kNN has also recently been tested for forecast updating by constructing a deterministic error prediction model (Akbari and Afshar, 2014). Similarly, it has been shown that model errors can be resampled using kNN, after explicitly accounting for input and parameter uncertainty, to generate uncertainty intervals (Sikorska et al., 2015). In this paper we extend the simplification of kNN resampling for uncertainty estimation. We present an application of the kNN method to generate residual uncertainty estimates for a predictand, using a fixed time series of input and fixed model parameters, and explore if this approach, being simpler many other uncertainty quantification approaches mentioned above, is a useful or even a better alternative.

To demonstrate its use, we employ a relatively simple configuration of kNN resampling to produce uncertainty intervals for hydrologic forecasting. The next section explains the method in more detail and describes the validation procedure, i.e. the performance indicators. Next, in section 3, the method is applied to two case studies, each with a different system response (discharge and water level). The performance of kNN uncertainty estimation as a function of forecast lead time is analysed in the first case study. Second case study is used to further validate the performance of kNN uncertainty estimation. Also, the influence of systematic bias in the hydrologic model on the uncertainty intervals generated by kNN search is explored in the second case study. For both case studies, performance indices of kNN resampling are compared to those of QR and UNEEC. And finally in section 4, we discuss the usability of kNN search as a postprocessor uncertainty estimator in hydrologic forecasting.





## 2 Method

### 2.1 kNN error model

The kNN residual uncertainty estimator can be seen as a zero[th] order local error quantile model built from a kNN search. Let us define a vector $\boldsymbol{v}$ in n-dimensional space of variables on which the residual uncertainty is assumed to be statistically dependent.

$$\boldsymbol{v} = [v^1, \ldots, v^n] \tag{1}$$

The cumulative probability distribution function C of residual errors at prediction time-step t conditioned on $\boldsymbol{v} = \boldsymbol{v}_t$ is defined as:

$$C_t(e|\boldsymbol{v} = \boldsymbol{v}_t) = P_t(E \leq e|\boldsymbol{v} = \boldsymbol{v}_t) \tag{2}$$

Where $P$ is the probability function and $E$ denotes the random variable for residual errors. Residual error is defined throughout this paper as the difference between the simulated values and the observed values for a hydrologic quantity system response $f$, like discharge or water level.

$$e = f_{\text{simluated}} - f_{\text{observed}} \tag{3}$$

We are making the assumption of stationarity in time so that past error distributions are representative of the future:

$$C_t(e|\boldsymbol{v} = \boldsymbol{v}_t) = C_p(e|\boldsymbol{v} = \boldsymbol{v}_t) \tag{4}$$

The subscript p denotes historical time series. Therefore $C_p$ is the cumulative distribution function of residual errors from the past. In Eq. (4), $C_p$ is being conditioned to the input variable vector at time t. Nevertheless, as we only have single realizations of the error variable $E$ for each historical point, we relax the constraint of $\boldsymbol{v} = \boldsymbol{v}_t$. Instead, we assume that the nearby neighbours of $\boldsymbol{v}_t$ in n-dimensional space will have a similar probability distribution of errors as $\boldsymbol{v}_t$ and that these historical errors are samples from $C_p(e|\boldsymbol{v} = \boldsymbol{v}_t)$. An empirical probability distribution can thus be constructed using the kNN historical errors:

$$C_t(e_t|\boldsymbol{v} = \boldsymbol{v}_t) \approx C_p(e|\boldsymbol{r}_p \leq \boldsymbol{r}_k) \tag{5}$$

where $\boldsymbol{r}_p$ is the Euclidean distance in n-dimensional space of input variables.

$$\boldsymbol{r}_p = |\boldsymbol{v}_p - \boldsymbol{v}_t| = \sqrt{\left[\sum_{i=1}^{n}(v_p^i - v_t^i)^2\right]} \tag{6}$$

$\boldsymbol{v}_p$ is the input variable vector of the past data point in the cloud of such past data points $\boldsymbol{v}$ (Figure 1)and $\boldsymbol{r}_k$ is the distance to the $k^{th}$ nearest neighbour of $\boldsymbol{v}_t$. Choice of the input variable vector is a problem in itself since it should include only the most relevant variables that determine the forecast uncertainty. In this study, the input variable vector is chosen based on correlation between the candidate variables and the past errors. If the correlation between the error time series and a particular candidate variable is relatively high, then it can be included in the input variable vector space. Other, more





sophisticated methods involving the mutual information can be used as well (Fernando et al., 2009). This will be exemplified in the case studies described in the next section. To represent relative importance of input variables used in the search, dimensions of the input variable vector space can be suitably weighted in. Also, the model-based methods can be used when the models are built for each candidate input variables set considered and the choice is made based on their relative performance. These, however, were not explored in this study; it rather focused only on the usability of k nearest neighbour search in its most basic implementation for uncertainty quantification. To level variables with different magnitudes they are normalized. If $\sigma_i$ represents the standard deviation of input variable I calculated using the past data, then:

$$\boldsymbol{r}_p = \sqrt{\left[\sum_{i=1}^{n} \frac{(v_p^i - v_t^i)^2}{\sigma_i^2}\right]} \qquad (7)$$

Once, the input variable vector space is decided, the probability of non-exceedance of a forecast error is calculated empirically by sampling from the conditional error distribution:

$$C_t(e_t | \boldsymbol{v} = \boldsymbol{v}_t) \approx C_p(e | \boldsymbol{r}_p \le \boldsymbol{r}_k) = j/k \qquad (8)$$

where $j$ is the rank of value e (for which the probability of non-exceedance is being computed) in the ascending array of $k$ error values. The kNN search is thus employed to generate a sample and to build an empirical error distribution for this predictive uncertainty quantification. As the configuration kNN used in this research generates residual error quantiles, which capture the mismatch between measurement values and simulated values, the uncertainty in observational data is not considered. The generated quantiles are aimed to capture the measured system response and do not attempt to capture the true response of the hydrologic system.

As one would expect, due to the nature of our sampling approximation (Eq. (8)), the number of nearest neighbours, k, will affect the empirical conditional probability distribution of errors. If k is large, many data points that are quite distant from $\boldsymbol{v}_t$ (Figure 1) will be selected and the conditioning on the current forecast situation will not be valid. Large values of k will thus yield error distributions resembling the marginal error distribution. If k is small, the set of k errors will be small and subject to sampling error, so this set will not adequately represent the uncertainty distribution at $\boldsymbol{v}_t$. For improved performance, the value of k can be subject to optimisation of some cost function: the optimal value of k could be the one that enables a reasonable estimate of the uncertainty quantiles and additionally we may require that the sensitivity of the error distribution to k is small. In this study, we carry out such optimization using quite a simple heuristic guideline - the value of k is varied until the probability distribution of errors stabilizes and becomes less sensitive on the value of k for a few model predictions. This will be demonstrated by an example in the case studies. The choice of this relatively simple procedure for error quantile generation using kNN resampling is a reasonable starting point to assess its potential for residual uncertainty. This study explores the potential of uncertainty estimation using kNN in as simple a way as possible. And then compare its performance to two other residual uncertainty estimators. More advanced application of kNN, for example using fuzzy



weights and kNN sampling to assign prediction intervals (Shrestha and Solomatine, 2008) or through explicit consideration of uncertainty in parameter and input by sampling them from their distributions, has been successfully shown (Sikorska et al., 2015).

5    To summarize, the steps for uncertainty quantification using kNN resampling are as follows:

1. Compose the input variable vector space ($v$) on which uncertainty will be conditioned. Correlation analysis can help find the most relevant variables.

2. Set the number of neighbours k (It can be identified by optimization as well).

3. For a forecast at prediction time-step t, identify the set of k nearest neighbours to the input vector $v_t$. This set
10      represents the hindcasts (forecasts in retrospect) most similar to $v_t$.

4. Use the residual errors from these k points to build an empirical error distribution for the forecast at time-step t.

5. Finally, identify the errors corresponding to the required quantiles (probabilities of non-exceedance) from this empirical distribution (In this paper, we use 5-95% and 25-75% quantiles).

## 2.2 Validation methods

Two statistical measures have been employed in this study to check the effectiveness of uncertainty estimation techniques, namely Prediction Interval Coverage Probability ($\text{PICP}_{\text{PI}}$) and the Mean Prediction Interval ($\text{MPI}_{\text{PI}}$) (see, e.g. Shrestha and Solomatine 2008; Dogulu et al., 2015). $\text{PICP}_{\text{PI}}$ represents the percentage of observations ($C$) covered by a prediction interval (PI) corresponding to a certain probability of occurrence (in our case 90% and 50%).

$$\text{PICP}_{\text{PI}} = \frac{N_{\text{in}}}{N_{\text{obs}}} \times 100\% \tag{9}$$

20 where $N_{\text{in}}$ is the number of observations located within PI and $N_{\text{obs}}$ is the total number of observations. These metrics are calculated using the following equations:

$$\text{PICP}_{90} = \frac{1}{n}\sum_{i=1}^{n} C_{90} \times 100\% \tag{10}$$

$$C_{90} = \begin{cases} 1, & \text{if } q_{i,0.05} \leq q_i \leq q_{i,0.95} \\ 0, & \text{else} \end{cases} \tag{11}$$

$$\text{PICP}_{50} = \frac{1}{n}\sum_{i=1}^{n} C_{50} \times 100\% \tag{12}$$



$$C_{50} = \begin{cases} 1, & \text{if } q_{i,0.25} \le q_i \le q_{i,0.75} \\ 0, & \text{else} \end{cases} \qquad (13)$$

where $q_{i,0.95}$ and $q_{i,0.05}$ are values with 95% and 5% probability of non-exceedance at time i. Thus the region bound within these two values will have a confidence interval of 90%. Similarly, $q_{i,0.75}$ and $q_{i,0.25}$ represent the boundaries for 50% *C*. The MPI is the average width of the confidence intervals corresponding to a particular probability. It is a measure of the magnitude of uncertainty.

$$\text{MPI}_{90} = \frac{1}{n} \sum_{i=1}^{n} (q_{i,0.95} - q_{i,0.05}), \qquad \text{MPI}_{50} = \frac{1}{n} \sum_{i=1}^{n} (q_{i,0.75} - q_{i,0.25}) \qquad (14)$$

There have been discussions whether an isolated verification index can capture all the aspects that make a probabilistic forecast good or bad (Laio and Tamea, 2007). The choice of a verification index for an uncertainty estimation technique should also be dependent on the purpose of hydrologic forecast. For example, Coccia and Todini (2011) evaluate the performance of Model Conditional Processor for flood forecasting using the predicted and observed probability of exceedance over a threshold. Also, in their study predicted error quantiles are compared to observed error quantiles. López

López et al. (2014) and Dogulu et al. (2015) use PICP and MPI, among other verification measures, to access the performance of QR and UNEEC. This study will limit the comparison of kNN resampling with other techniques to PICP and MPI only, which give a reasonable assessment of performance. Nevertheless, it does not preclude the possibility that the uncertainty estimation techniques perform differently if evaluated using other indices.

**3 Case studies**

The performance of kNN resampling was evaluated by applying the technique to hydrological forecasting for several catchments in two different parts of England. The two case studies provide two different hydrologic conditions for testing and include different models for prediction. Also, different kinds of system responses are being predicted in the two case studies – water level and discharge. The accuracy of the quantified prediction intervals was deduced by using validation data

sets. Also, the first case study was used to evaluate the impact of changing lead time on uncertainty of hydrologic models and its quantification using kNN resampling.

**3.1 Upper Severn catchment**

**3.1.1 Catchment description**

Upper Severn region is located in the Midlands, UK (Figure 2). River Severn, with a total length of 354 km, is the longest

river in the UK. Its course acts as a geographic delineation between England and Wales, finally draining into the Bristol



Channel. The overall River Severn catchment area is 10,459 km$^2$. Around 2.3 million people live in this region. The area is predominantly rural, but there are also a number of highly urbanized parts. The area covering the upper reaches of River Severn, from its source on Plynlimon to its confluence with the River Perry upstream of Shrewsbury in Shropshire, is called the Upper Severn catchment. The Upper Severn catchment is predominantly hilly. It is dominated on the western edge by the

Cambrian Mountains and a section of the Snowdonia National Park (River Severn CFMP. EA, 2009).

The Severn catchment has a diverse geology. The headwaters of the River Severn rise on Silurian mudstones, siltstones and

grits and flow eastwards over these same rock formations.  These rock formations do not allow water to flow easily through them. Therefore they are classified as non-aquifers with only limited potential for groundwater abstraction. Further west, in the Middle Severn section, the River Severn encounters sandstones, which are classified as a major aquifer and are highly permeable, highly productive and able to support large groundwater abstractions (River Severn CFMP. EA, 2009). The climate of the Severn catchment is generally temperate, experiencing modest to high precipitation dependent on topography.

Welsh Mountains can receive over 2,500 mm of precipitation per annum, whereas the rest of the catchment receives rainfall similar to the UK average - less than 700 mm per annum. The test forecast locations used in this study are Llanerfyl, Llanyblodwel and Yeaton. Table 1 lists the basin and hydrological information for these subcatchments (López López et al., 2014).

**3.1.2 Experimental setup**

Flood forecasting system for River Severn is organized in a sequential manner, being composed of a number of separate systems that are effectively linked. This forecasting system works with a high degree of automation and efforts have been made to involve minimum amount of human intervention. . UK Environment Agency uses Midlands Flood Forecasting System (MFFS) to do flood forecasting and to help in warning operation. MFFS in turn is based on Delft-FEWS, Flood

Early Warning System) platform (Werner et al., 2013). Within MFFS, there are lumped numerical models for rainfall-runoff (MCRM; Bailey and Dobson, 1981) and models for hydrologic (DODO; Wallingford, 1994) and hydrodynamic routing (ISIS; Wallingford, 1997). The rainfall input for MFFS is acquired from ground measurements via rain gauges, from radar measurements or from numerical weather prediction data. MFFS predict ahead in time the response of the Upper Severn subcatchments but, as expected, the quality of forecast deteriorates with increasing lead time.

To do uncertainty analysis for MFFS, hindcasting or reforecasting is done and then simulations are compared to data. All the input time series used for hindcasting is taken from measured data. In this study, the reforecasting period was kept equal to that in the study of (López López et al., 2014) and (Dogulu et al., 2015).The chosen period is from 1 January 2006 through 7





March 2013. Data in the period till 6 March 2007 is used for the model spin up. The remaining period is used for the calibration and validation of the uncertainty estimation techniques. Forecasts are made on a 12 hourly basis – at 8:00 and 20:00 daily, up to a lead time of 48 hr. kNN resampling was applied for forecasts at 10 different lead times: 1, 3, 6, 9, 12, 18, 30, 36, 42, and 48 hr. To choose an input variable vector for kNN resampling, correlation analysis was done between

residual error and contenders for input variable vector space, namely simulated water level ($H^{sim}$), measured water level ($H^{obs}$) and residual error ($E^{obs}$) from various time steps $t$. The analysis was done to assist in a manual selection of input variable vectors. The correlation between residual error and water level reduces fast with time lag between the two time series. Therefore it is enough to choose relatively simple and small dimensional input variable vector spaces. For lead times, $l$, up to 6 hours we chose:

$$\boldsymbol{v} = [H_t^{sim}, H_{t-1}^{obs}, e_{t-1}^{obs}] \tag{15}$$

For higher lead times, uncertainty has only been conditioned on $H_t^{sim}$ as the residual error becomes less and less correlated with variable values at measured several hours behind the prediction time.

$$\boldsymbol{v} = [H_t^{sim}] \tag{16}$$

Ninety nine values of residual errors were sampled from the nearest neighbourhood to generate an empirical distribution at each prediction step. This allowed us to get the 'resolution' of 1 percentile in the generated empirical distribution. To develop confidence in the chosen value of k, we checked for a few prediction steps how sensitive the generated empirical

distribution is to the value of k. Four different instances of $\boldsymbol{v_t}$ were chosen. Each instance represents a prediction step in the input variable vector space (the red circle in Figure 1), with different hydrologic conditions. The plots of the cumulative mean square difference between pdfs of varying k were generated. Cumulative mean square difference (Eq. (17)) just serves as an index to show how much the empirical pdfs change with changing k. We get a decreasing slope with increasing k. It shows that the pdfs become almost identical for values of k around 100. If $P_{k_i}(e)$ is the probability density for a residual

error e calculated through $k_i$ nearest neighbours using kNN resampling, for probability functions corresponding to discrete bin size $\Delta e$, the cumulative difference is defined as:

$$\text{Cumulative Difference} = \sum_{k_i=10}^{k_i=k} \sum_{\text{first e bin}}^{\text{last e bin}} [\Delta e \cdot P_{k_i+1}(e) - \Delta e \cdot P_{k_{i-1}}(e)]^2 \tag{17}$$

The various values of $k_i$ that were tested are -10, 30, 50, 70, 90, 100, 110, 130 and 150. Using the information from Figure 3, a value of k = 99 does not seem to be heavily effected by sampling errors. Nevertheless, it is not a mathematically calibrated value of k and therefore is likely to be sub-optimal. However, it should still be able to provide reasonably

representative samples from the error distribution, as is suggested by Figure 3.

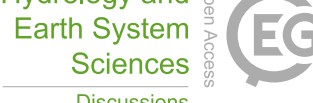



### 3.1.3 Results

Figure 4 shows two hydrographs for the same event, where model predictions were made at different lead times. From the graph of lead time 48 hr it is evident that the error quantiles that kNN resampling produces are not forced to have zero mean. Therefore the model prediction can sometimes lie outside the predicted quantiles. This is because kNN resampling learns

from past instances where the model has consistently under or overpredicted the flow, so it corrects for this bias. The hydrographs capture the low flows and the peaks well. It can also be seen that for high flows the errors are usually higher than for medium and low flows. The residual error distribution is thus heteroscedastic, i.e. the variance depends on the magnitude of the predicted flow. The autocorrelation can be checked by plotting errors versus time. Whereas performance of an error model with regard to heteroscedasticity can be estimated by plotting reliability diagrams for different magnitudes of

flow, which would mean different water levels in this case.

The plotting of error time series for various lead times shows some recurring trends across all the three subcatchments. The errors are small for small lead time forecasts and the spread of error time series increases with increasing lead time. Moreover, the errors do not look to be autocorrelated for smaller lead times, whereas for the higher lead times

autocorrelation becomes more prominent. This can be ascribed to the memory of the hydrologic system. If the system response is higher than what the model simulates for a particular lead time, then the system response is likely to be higher for the next time step as well. As the errors become larger, they tend to lose their independence property. This is captured by the error samples generated by kNN resampling as well. The rate at which autocorrelation deteriorates for observed residual errors corresponds well to the kNN resampling error samples' autocorrelation (Figure 5).  It can be seen that kNN

resampling preserves the autocorrelation in the error time series without using an autoregressive model.

To check the performance of kNN resampling for various flow magnitudes, the simulation values were divided into low and high flows - the lowest and the highest 10 percent of water levels simulated in the validation phase respectively. The reliability diagrams (Figure 6) shows that the overall performance of error quantiles for all water levels is good for low and

medium lead times. The reliability decreases with high lead times (24 hr and above). The reliability plots show that kNN resampling performs better for high flows compared to lows flows, even for higher lead times. For lows flows and high lead times, the forecast probability of non-exceedance is higher than the observed relative frequency. Nevertheless, from 0.90 probability of non-exceedance and above, the reliability curve comes back to the desired $45^0$ line.  For flood forecasting it is important to model the high and medium flows well. kNN resampling delivers quite reliable quantiles for such flow regimes.

The deteriorating model performance with higher lead times gets reflected in the performance of kNN resampling quintiles as well.

To assess the performance of kNN resampling relative to other established post processor uncertainty estimation techniques, comparisons with QR and UNEEC have been carried out. The results for QR have been taken from  López López et al.





(2014) and the results for UNEEC – from Dogulu et al. (2015). QR results for uncertainty estimation were available for the all the lead times as done using kNN resampling, and from UNEEC – only for lead times of 1, 3, 6, 9, 12, and 24 hr. Values of PICP and MPI are shown in Figure 7, together with results from UNEEC and QR. As expected, the mean prediction interval (MPI) of all the uncertainty estimation techniques increases with increasing lead time. Comparison between kNN resampling and QR has been made for 3 locations, 10 lead times in the validation period. Model simulations were run two times each day. Verification indices for uncertainty analysis were calculated separately for each lead time and each location. Considering 90% and 50% quantile as two prediction intervals, this allowed for the evaluation of PICP and MPI 60 times (Figure 7). kNN resampling has higher PICP in 67% of the cases and a smaller MPI for 73% of the cases. A comparison between kNN resampling and UNEEC was made for 3 locations, only 5 lead times for the validation. For each location and each lead time, the 90% and 50% quantiles were generated, which allowed for the evaluation of PICP and MPI 30 times (Figure 7). The PICP of kNN resampling is higher in 60% of the cases and the MPI is smaller in 36% cases. Based on these results we concluded that, for this case study, kNN resampling generally produces narrower confidence bands and provides a better coverage of the probability distribution than the other methods in the majority of forecasts, especially showing better performance for the larger lead times.

## 3.2 River Brue

### 3.2.1 Catchment description

River Brue, located in the south west of England, has a history of severe flooding. The test forecast location used in this study is Lovington, where the upstream catchment area is 135 km$^2$ (Figure 8). The catchment is predominantly rural and the soil consists of clay and sand. This kind of soil and the modest relief give rise to a slowly responsive flow regime. The mean annual rainfall in the catchment is 867 mm, the mean river flow is 1.92 m$^3$/s. and has a maximum flow of 39.58 m$^3$/s. This catchment has been extensively used for research on weather radar, quantitative precipitation forecasting and hydrologic modelling.

### 3.2.2 Experimental setup

For Brue catchment the simplified version of the HBV rainfall-runoff model has been used (Bergström, 1976). HBV-96 model is a lumped conceptual model (Lindström et al., 1997). Like most other conceptual models, HBV consists of subroutines for snow accumulation and melt, soil moisture accounting and surface runoff, and employs a simple routing scheme. The input for the HBV model consists of precipitation (basin average), air temperature and potential evapotranspiration (estimated by modified Penmann method using automatic weather data available). Historical input data is available for a period of 1994-1996. Predictions are only made for 1 hr lead time. Uncertainty analysis is done for a chosen





period from 24th June 1994 through 31st May 1996. Hindcasts were made on a daily basis, using a warm state from a historical run. The hindcasts were split into calibration and validation set at 24$^{th}$ June 1995 for the uncertainty estimation techniques. The calibration data set was used to calibrate (train) UNEEC and QR, whereas kNN resampling was allowed to learn only from this same calibration data set to estimate prediction intervals for the vectors from the validation data set.

Each of the two data sets represents almost a full year of observations. The input variable vector was chosen based on the results of correlation analysis. To make an appropriate comparison, the same variables have been used for kNN resampling as used for UNEEC in (Dogulu et al., 2015). The input variable vector is:

$$\boldsymbol{v} = [R_{t-8}^{obs}, R_{t-9}^{obs}, R_{t-10}^{obs}, Q_{t-1}^{obs}, Q_{t-2}^{obs}, Q_{t-3}^{obs}, e_{t-1}^{obs}, e_{t-1}^{obs}] \qquad (18)$$

where $R$ is the effective rainfall, $Q$ is the discharge and $e$ is the residual error. Considering t as the prediction time, then the subscript of the various input variables represents the time and the superscript *obs* means they are observed values. The number of nearest neighbours was chosen to be 99, after a similar manual calibration procedure as for the Upper Severn case study. Uncertainty analysis was done for a calibrated HBV model as well as a model with a unit systematic bias. The bias was introduced to the simulation results of the calibrated model by simple addition. The aim of a biased model for

uncertainty quantification using kNN resampling is to assess the performance of kNN resampling when the residuals are not zero mean.

### 3.2.3 Results

kNN resampling was applied to a single historical simulation and compared to observations. The simulated hydrograph for

two events with 50% and 90% prediction intervals are shown in Figure 9. The residual distribution of kNN resampling is generally non-zero mean. Therefore we see that the prediction intervals may sometimes deviate from the deterministic model prediction quite significantly. The ability of kNN resampling to search for similar hydrologic conditions, like rainfall, and discharge in the past, and learn from the residuals, allows it to make more representative error distributions. For example, in Figure 9, the falling limb of hydrograph in event 2 shows that the prediction band generated by kNN resampling captures the

observed flow almost perfectly, even though the model shows a noticeable mismatch with the measurements. This can be explained by considering the history of errors that the model made during such hydrologic conditions in the past. And as kNN resampling learns that the model consistently underestimates in such cases, the corresponding error distribution corrects for this bias. The results of the PICP and MPI are shown in table 2 together with results from UNEEC and QR (Dogulu et al., 2015). As can be seen from the table, kNN resampling performs better than the UNEEC and QR for this case study. The

prediction intervals generated by kNN resampling are smaller, compared to the other two uncertainty estimation techniques,



while the coverage probability is similar. It indicates that kNN resampling is able to learn well from past data and condition the probability of residual errors well.

Apart from evaluating the usability of kNN resampling for calibrated models, the performance of kNN resampling quantiles generated by kNN resampling for a model with systematic bias was also checked. Figure 10 shows that the performance of kNN resampling does not diminish under systematic bias. The reliability of the generated quantiles remains almost unfazed. As a systematic bias will not affect the autocorrelation structure of the residual errors, the autocorrelation of error samples generated through kNN resampling also remains unchanged. Nevertheless, we see a shift in the mean of the sample time series, which is roughly equal to unity. The reliability of quantiles generated using kNN resampling for high flows (highest 10% in the validation period) is as good as for all flows. Thus kNN resampling maintains its accuracy for flow regimes that are of interest in flood forecasting, like it did for Upper Severn catchments. The invariance of kNN resampling performance to model bias and high predictand magnitudes of makes it a robust post processor uncertainty estimation technique.

## 4 Discussion and conclusions

The application of kNN resampling to two cases studies shows that the forecast uncertainty intervals are relatively narrow and still capture the observations well. The expected increase of uncertainty for longer lead times is also reproduced well and the probability coverage of kNN resampling remains good as verified from historical observations. This is in accordance with previous research (Sikorska et al., 2015). The error samples generated by kNN resampling reproduce two important characteristics of residual errors in hydrologic models namely autocorrelation and heteroscedasticity. Also, for applications to flood modelling, the high flows are most important and the uncertainty quantification by kNN resampling for both case studies, Upper Severn and Brue, shows reasonable reliability for this high flow regime. Moreover, it is shown that the technique is generally robust to the performance of the underlying deterministic model. If the model has systematic biases, kNN resampling learns from the past errors of the model and recreates the systematic bias in the empirical error distributions mean, thus maintaining the performance of prediction intervals. Our results on systematic error correction by kNN resampling substantiate the findings from previous research on forecast updating using kNN (Akbari and Afshar, 2014). These findings from this study are confirmed by two quantitative indicators of forecast reliability. The comparison of kNN resampling uncertainty estimates to those generated by QR and UNEEC show that the mean prediction intervals (MPI) generated by kNN resampling are generally smaller. kNN resampling is generally able to capture the expected ratio of observations within its intervals (PICP) most of the times, or at least be close to the expected value.

As all data-driven methods, the applicability of kNN resampling depends on the availability of sufficiently long and representative historical forecasts and observations. The historical series should include several occurrences of forecasting



situations that are similar to the current situation. In extreme cases, the kind of kNN search proposed here will select the most similar historical situations which may or may not be representative of the current situation. In contrast to the methods like QR and UNEEC that build explicit predictive regression models which are able to extrapolate for the data which is beyond the limits of the calibration (training set), kNN resampling does not extrapolate. This could be seen as a

disadvantage. On the other hand, however, the extrapolation that is done by regression techniques could be also seen as doubtful. It is not a given that the most extreme historical situations are less representative for the uncertainty of an extremely high flow than an extrapolated result. The results of both case studies in this paper show that kNN resampling has a good reliability for the highest values in the validation set. Due to the non-parametric nature of kNN resampling, the increasing variance of residual errors for higher values of predictand is adequately taken into account.

As kNN resampling, like other post processors, learns about the residual error process from the past, the historical records should be representative of the current forecast conditions. In changing conditions, this may not be true. Changing conditions may be caused, for example due to climate change or more local changes in the catchment like deforestation, dam building etc. This is a common problem for all data-driven statistical estimators and not unique to kNN resampling. Care needs to be

taken to use data time series which do not outrightly violate the assumptions regarding the invariance of catchment and climate.

One of the few calibration parameters of kNN resampling is the number of nearest neighbours k. In this study, k has been chosen by a simple heuristic technique. For optimal performance, it would be advisable to calibrate k for each application in

a more systematic way. We expect that the optimal value of k will depend on the length of the historical data series and on the uncertainty quantiles of interest. Also, in this research the input variable vector has been chosen by correlation analysis. It can be recommended to use more sophisticated procedures for real life applications. It is foreseen that improvements in performance can possibly be achieved by seeking a better set of input variables for each forecast location and lead time of interest.

In conclusion, kNN resampling can be considered as a relatively simple machine learning technique to predict hydrologic residual uncertainty. The errors from the similar hydrologic conditions in the past are used as samples for the residual error probability distribution and the samples are collected by a k nearest neighbour search. The application of this technique to case studies Brue and Upper Severn catchments has shown promising results. In comparison to many other data driven

techniques, kNN resampling has the advantage of avoiding assumptions about the nature of the residual error distribution: the instance-based learning approach is non-parametric and non-regressive and requires little calibration. The method was shown to be able to quantify hydrologic uncertainty to an accuracy that is comparable to other techniques like QR and UNEEC. Given the relatively small effort in setting up the method, the performance of kNN resampling in uncertainty quantification is more than acceptable when compared to other post processor error models.





**User interface**

A website has been developed as part of this research to help generate uncertainty estimation intervals using kNN resampling for a given time series of predictions. Address: www.modeluncertainty.com

**Acknowledgements**

The authors would like to acknowledge Bonneville Power Administration, Portland, USA, for supporting this research. The UK Environment Agency is acknowledged for provision of the data for the case studies described in this manuscript. Many thanks to Nilay Dogulu, Patricia López López, and Marijn Swenne for their help during the course of this research. We are thankful to Sven Eggimann and H. Badger for helping structure the paper. Part of this study was supported by the EC FP7
project WeSenseIt (Citizen Observatory of Water), grant agreement no 308429 and project QUICS (Quantifying Uncertainty in Integrated Catchment Studies), grant agreement no. 607000.

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




**Table 1: Basin information for Upper Severn catchments**

5 **(EA, 2013 and Marsh, T. J. and Hannaford, J., 2008)**

| Catchment | Area (km$^2$) | Mean Annual Rain (mm) | Mean Flow (m$^3$/s) | Max Water level (m) |
|---|---|---|---|---|
| Llanerfyl | 125 | 1077 | >10 | 3.59 |
| Llanyblodwel | 229 | 1267 | 6.58 | 2.68 |
| Yeaton | 180.8 | 767 | 1.6 | 1.13 |





5 **Table 2: Performance of various uncertainty estimation techniques for Brue catchment**

|  | PICP (Expected 90%) | | | MPI (m$^3$/s) | | |
| --- | --- | --- | --- | --- | --- | --- |
|  | UNEEC | QR | kNN | UNEEC | QR | kNN |
| Calibration | 91.19 | 90.00 | 95.11 | 1.58 | 1.69 | 0.49 |
| Validation | 88.29 | 82.33 | 92.15 | 1.37 | 1.39 | 0.39 |





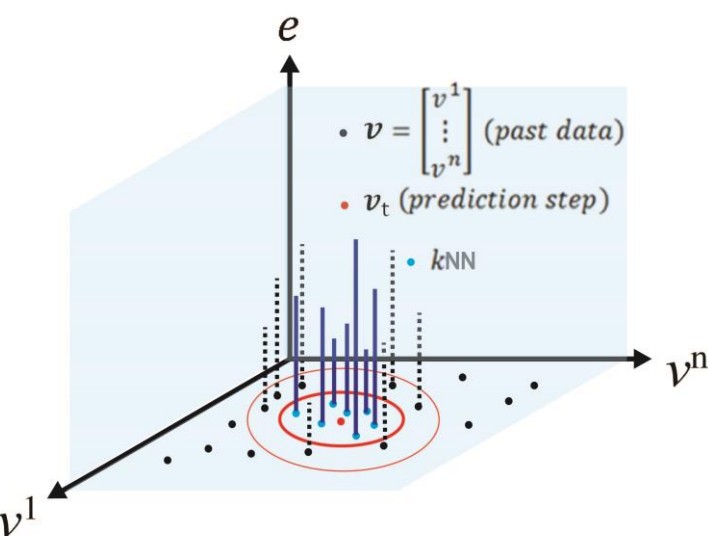

5    **Figure 1. Dependence of error samples on the value of k. For larger values of k, points are at a greater distance from**
$v_\mathrm{t}$ **(the prediction step), thus compromising the conditioning of the residual error probability distribution on** $v_\mathrm{t}$ **(Eq.**
**(5)).**





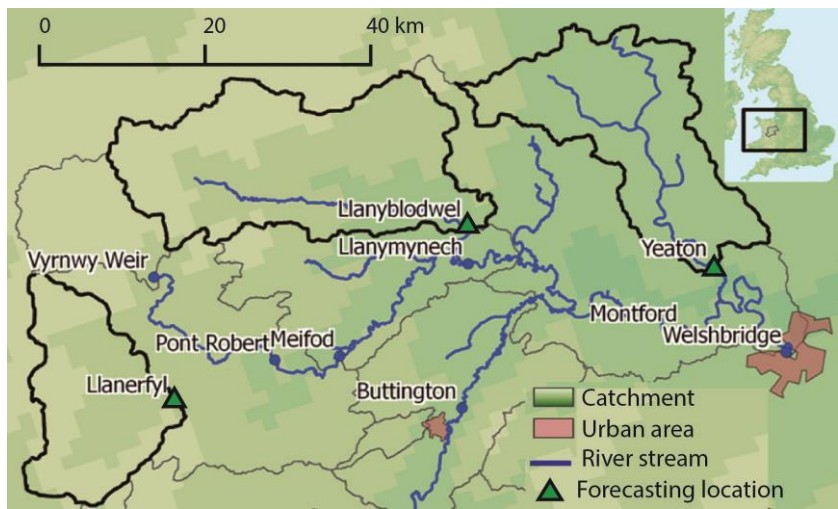

**Figure 2. Upper Severn catchments with gauging stations (From López López et al., 2014)**



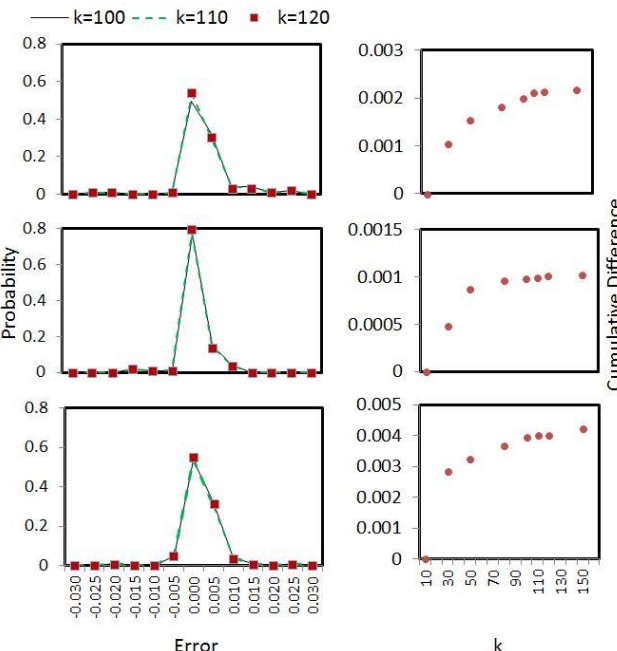

**Figure 3.** **Dependence of residual error probability function on the value of k for three didactic values of $v_t$ (each row). The probability is computed for error bins of size 0.005 units each. The graphs show that for k from around 90 to 120, the corresponding empirical error distributions become almost identical.**





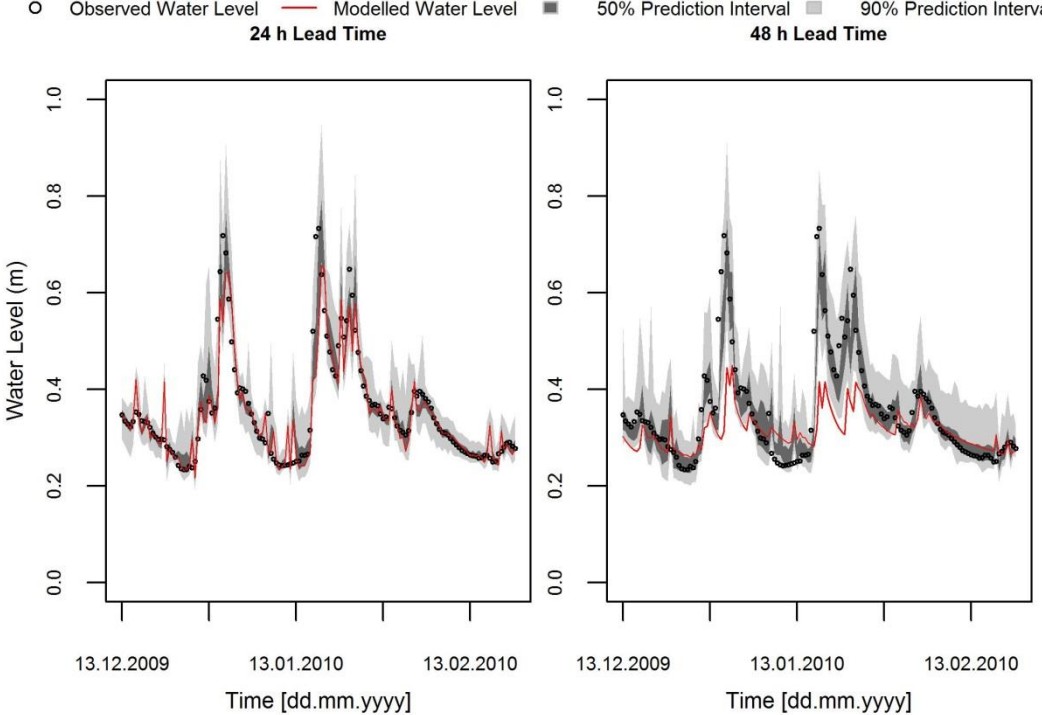

**Figure 4.** Prediction intervals for Yeaton catchment using kNN resampling. The hydrographs are shown for the two different lead times. 50% prediction interval is the interval between 25% and 75% quantiles of residual error, and 90% quantile is the interval between 5% and 95% quantiles. The reporting time interval is 12 hours.




**Figure 5. Plots of error samples and their autocorrelation (ACF). The error time series generated using kNN resampling are in red. Black circles represent the observed errors, i.e. obtained after measuring water level and comparing it to simulated water level. M stands for measured and S for simulated. The lead time for each row of plots is given in blue.**







**Figure 6.** **Reliability diagram from Upper Severn catchments for high, low and all flows. (Llanerfyl – blue, Llanyblodwel – green, Yeaton - red).**



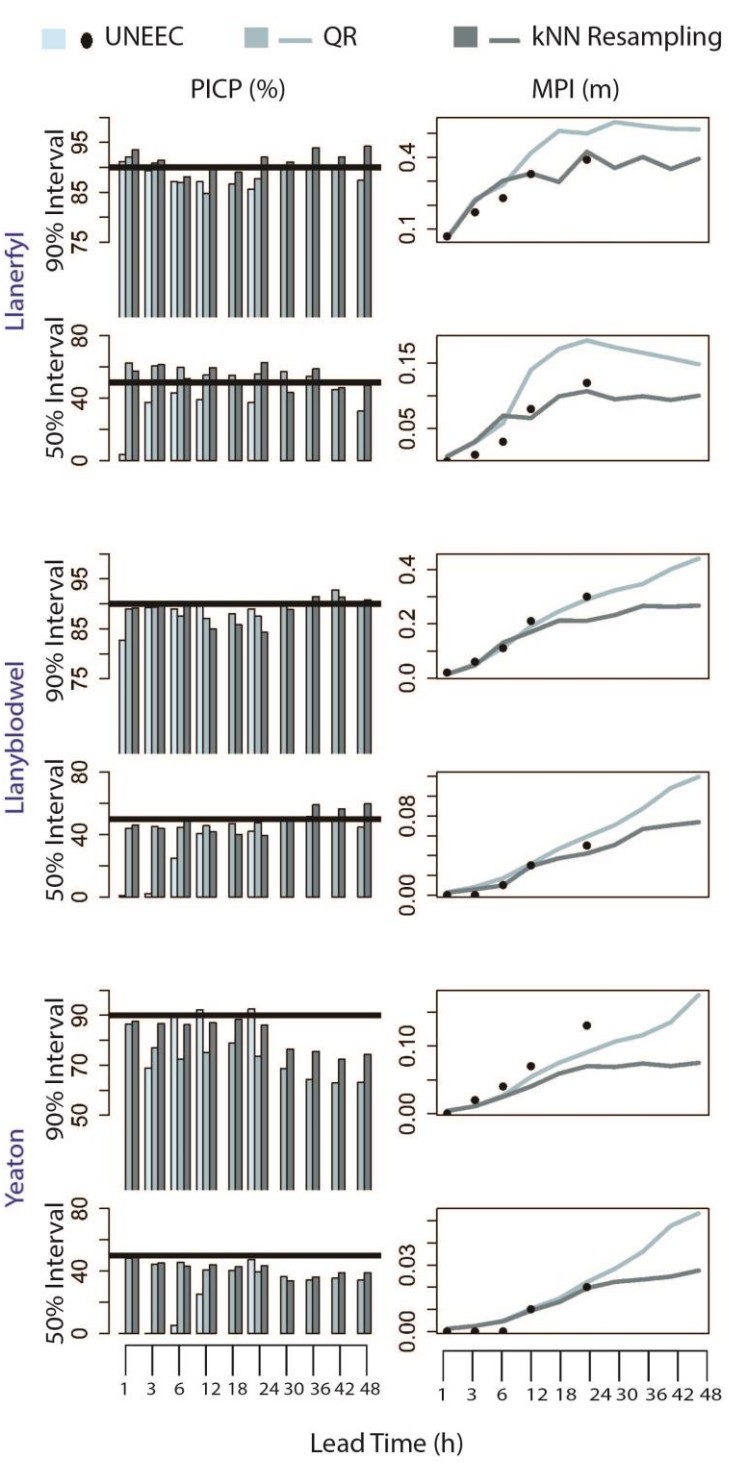

**Figure 7. PICP and MPI comparison for Upper Severn catchments.**





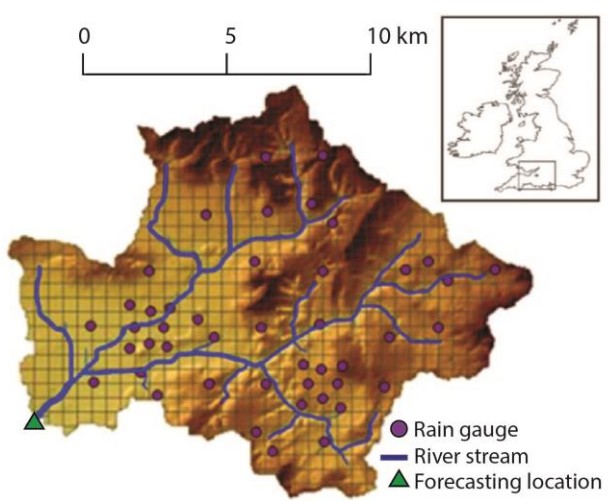

**Figure 8. Brue catchment (from Shrestha and Solomatine, 2008)**





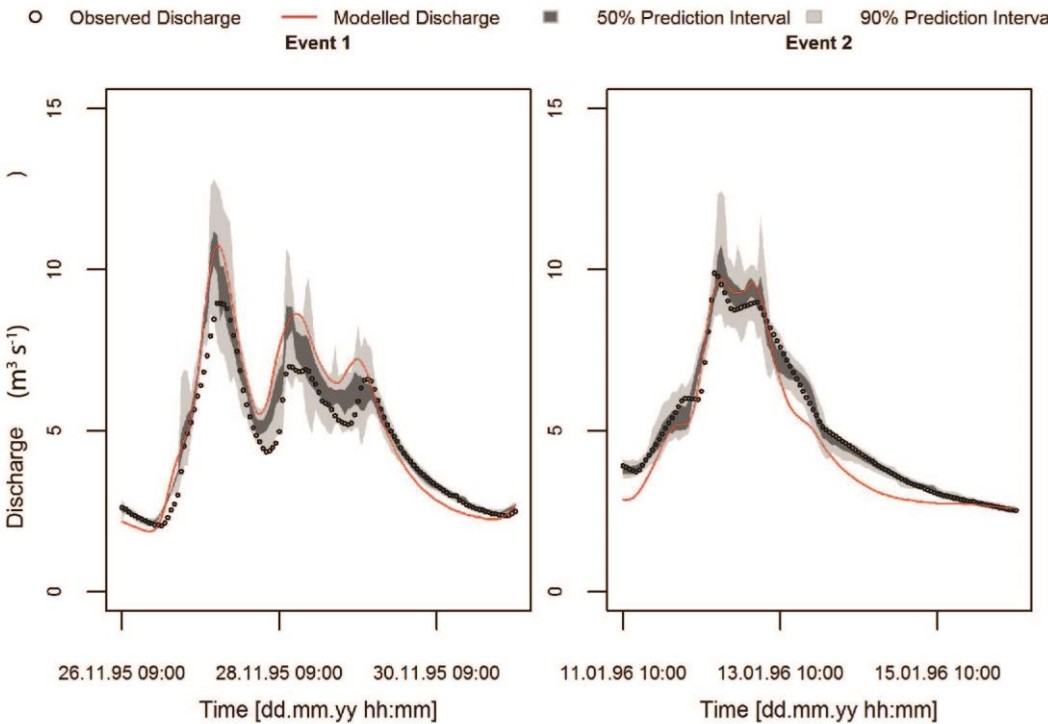

**Figure 9. 50% and 90% prediction intervals for Brue catchment using kNN resampling. The hydrographs are shown for two different events. (50% prediction interval is the interval between 25% and 75% quantiles of residual error, and 90% quantile is the interval between 5% and 95% quantiles.)**



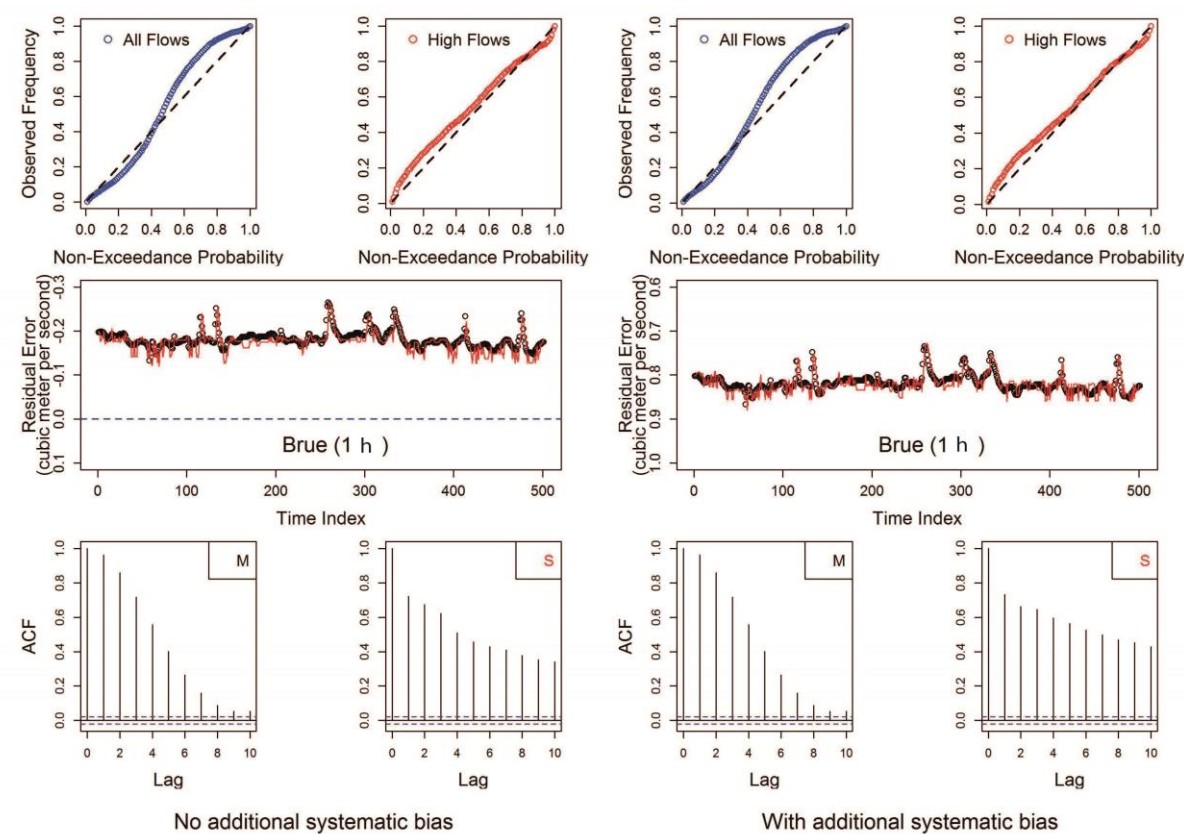

5 **Figure 10.** **Effect on reliability of quantiles and autocorrelation of error samples on adding a systematic bias to the model artificially. kNN samples are plotted in red and observed errors in black circles. M stands for measured and S for simulated.**