# Peer review of "Residual uncertainty estimation using instance-based learning with applications to hydrologic forecasting"

_Hydrology and Earth System Sciences, 2017_

## Referee Comment (RC1) · J. Matos (Referee) · 9 Apr 2017

Review comment on the paper entitled "*Residual uncertainty estimation using instance-based learning with applications to hydrologic forecasting*" by Omar Wani, Joost V.L. Beckers, Albrecht H. Weerts, and Dimitri P. Solomatine, submitted to Hydrology and Earth System Sciences on the 16[th] of March 2017.

**General comments**

The manuscript proposes a non-parametric method to estimate the uncertainty associated with residuals of deterministic models applied to hydrologic forecasting.

The method relies on the well-known k-nearest neighbors (kNN) technique, being simple both conceptually and in its application. It is compared to two other post-processor techniques – Quantile Regression (QR) and Uncertainty Estimation based on local Errors and Clustering (UNEEC) – over two test cases in the UK.

The paper is well written and clear. Also, the title and abstract fit the paper's contents. Most relevantly, in my opinion the paper addresses a relevant scientific question related to finding non-parametric methods of simple and broad application that allow accurate estimations of predictive uncertainty in hydrology, a question which falls into the scope of HESS.

The authors evidence knowledge of the topic and relevant past publications. The analyses that were undertaken are well described, as is the kNN method that is proposed. The results of the comparisons with both QR and UNEEC are clear and obviously benefit from the work that some of the authors did and published on those models.

I believe that the paper is a valuable contribution to the community and recommend its publication following the clarification of some aspects discussed in the specific comments.

**Specific comments**

The authors should adopt only one version of the terms "post-processor", "postprocessor", and "post processor".

The provided URL: [www.modeluncertainty.com](http://www.modeluncertainty.com) does not seem to work (tested on two days roughly one week apart).

The chosen measure of proximity was the Euclidean distance and the different variables on which the model is conditioned upon are normalized. Whereas in a model such as QR the importance of each variable is evaluated by the model, in kNN a variable's importance is defined *a priori* by its scale. The possibility of attributing different ways to input variables is mentioned, but not developed nor the target of a sensitivity analysis. I have doubts that kNN performs as well as QR (for example) if the informative inputs are mixed with less informative ones (and their weights are not properly adjusted). If so, the advantages in simplicity that kNN may have risk being offset by a more demanding input selection and preparation process.

The proposed methodology does not handle extreme events well, as possible future realizations are solely drawn from past observations. This is acknowledged by the authors, but I believe it should be emphasized sooner (e.g. already in the abstract) and more emphatically. While it is true that extrapolation poses a problem to many data-driven models, the intrinsic inability of kNN to perform

it at all cannot be seen other than as a disadvantage. That fact should be well-known by anyone who implements the approach in an operational setting.

Regarding validation metrics, it would be nice to add encompassing reliability assessment criteria such as the index α and the index ξ [e.g. *Renard et al.*, 2010].

It would be important to specify in the paper whether the inputs for kNN, UNEEC and QR are the same within each catchment. If this is not the case the authors should justify how may/do differences affect results and specify what part of the comparative performances is due to model structure (kNN, UNEEC and QR) and what part can be associated with the choice of input data.

The Mean Prediction Interval (MPI) results presented in Table 2 are striking. I do not believe that the discussion on why the results obtained by kNN are so strikingly better than UNEEC's or QR's justifies such a remarkable improvement and would very much like to understand the underlying reasons.

Related to the previous question, I would also like to see results on the performance of the kNN method when past error observations are not added to the input vector.

Please check the References for missing information (e.g. *Sikorska et al.*).

**Technical comments**
The technical comments are included below.

**References**

[revised manuscript text omitted]

---

## Referee Comment (RC2) · L. Raso (Referee) · 21 Apr 2017

**General comments**

The manuscript explores and discusses the application of k-Nearest Neighbors (kNN) method, a non-parametric machine learning technique, to estimate the predictive uncertainty in heteroschedastic streamflow forecasting.

The paper is clearly written. It comes completed of a internet website where a userfriendly interface makes application of kNN straightforward. The innovation is well framed in the recent literature on predictive uncertainty of heteroschedastic processes in hydrology, giving particular attention to comparable methods that estimates predictive uncertainty a posteriori. The authors clearly present advantages and limits of kNN with respect to other methods. Nonetheless, as already mentioned by the other referee, the limits of kNN in extrapolation, presently mentioned only in the conclusion, should deserve more emphasis.

The manuscript brings a valid and innovative contribution to its field, and I suggest its acceptance. There are two issues, however, that could contribute to make the case for this methodology in a more convincing way, and some minor issues that deserve at least to be mentioned.

The first main issue regards the selection of the k value, i.e. the number of data points considered similar to the instance to be estimated. Fixing k is a problem of kNN method. In general, when kNN is used for prediction, k is selected in order to maximize the predictive capacity, tested by a cross-validation on data. In the manuscript the criteria for selecting k is the stabilization of residuals probability distribution. Change in residuals distribution is quantified by the cumulative difference, defined at Equation (17). The reason why the stabilization of residuals distribution is a good criteria for fixing k is not clear. Moreover, this value is monotonic, hence it does not offer a clear-cut rule. The authors propose that k is to be selected when shape changes, but this rule, differently from what stated ad page 6 line 8, is not fitted to be used in an optimisation procedure.

The second issue regards the estimation of quantiles. kNN use the closest k values to build up an empirical distribution made of situations (i.e. data-points) similar to the "true" distribution that one intends to estimate. When kNN is applied for regression, the value to be predicted is the expected value, then the algorithm takes the average of k nearest data-points. In the proposed application instead, the empirical distribution is used to estimate some quantiles. Quantile estimation, however, has a different convergence rule than the expected value, particularly critical in estimating tails. Convergence rules of empirical distribution at quantiles of interest is well described in [1], chapter 21. Error in quantile estimation decreases with root square of k, and it is larger for quantile.

tiles close to 1 and 0. Using the 99th value from a set of 100 points as estimator of the 99th quantile may not be sufficient in guaranteeing sufficient convergency. Quantile estimation from empirical distribution introduce an error that must be be considered, or at least discussed.

I report here other smaller issues, worth to be mentioned in the manuscript.

In the discussion on verification index, the authors show that they are aware of the limits in using few indicators. The authors state that "PICP and MPI [...] give a reasonable assessment of performance". But this is not further explained. There are likely good reasons to select these indicators, but this should be better explained in the text, considering also that the application is about flood forecasting.

In Equation 7, variables are standardized one at a time, losing information about covariance. Why not considering variables as a multidimensional distribution, then using the covariance matrix to standardise? This would make use of the mutual information about variables in a more efficient way.

**Other comments**

Page 3, line 24: add "than" after simpler Page 5, line 12: "uncertainty in observational data is not considered", why can not it be included?

Page 8 line 23: remove extra dot.

Page 9 line 17: the adverb "just" looks like non necessary.

Page 10 line 11: The result description would be easier to follow if the reference to the figure was placed at the beginning of this paragraph (from line 19 to line 11).

[1] Van der Vaart, Aad W. Asymptotic statistics. Vol. 3. Cambridge university press,

**2000.**

---

## Author Comment (AC1) · 23 Jun 2017

Zurich, 23. June 2017

**HESSD MS No.: hess-2017-75 – Authors' Reply**

Dear Editor and Reviewers,

Please find below our revised manuscript according to the reviewers' feedback. Apart from addressing the minor comments, as a major modification, we have added new analysis to the study.

In this version one can find:

1) Additional simulation experiments for Brue catchment, where uncertainty analysis is done using three different input variable vectors and two values of k (99 and 199). This will help establish some idea of the dependence of kNN resampling performance on the choice of search space and k value.

2) Revised sentences to add proper caveats and qualifications to the performance of kNN resampling. The limitations have been stated explicitly.

We are grateful for the valuable feedback of both reviewers and the editor. We thank you for directing us towards potential improvements.

We hope that the revised manuscript meets the expectations and are thankful again for the effort of all parties involved in this review process.

Sincerely,

Omar Wani *(on behalf of the authors)*

*Please Note:* *The reference of all line numbers refers to the revised manuscript without marked changes.*

**Comment Reviewer 1 (J. Matos)**

General comments
The manuscript proposes a non-parametric method to estimate the uncertainty associated with residuals of deterministic models applied to hydrologic forecasting.
The method relies on the well-known k-nearest neighbors (kNN) technique, being simple both conceptually and in its application. It is compared to two other post-processor techniques – Quantile Regression (QR) and Uncertainty Estimation based on local Errors and Clustering (UNEEC) – over two test cases in the UK. The paper is well written and clear. Also, the title and abstract fit the paper's contents. Most relevantly, in my opinion the paper addresses a relevant scientific question related to finding nonparametric methods of simple and broad application that allow accurate estimations of predictive uncertainty in hydrology, a question which falls into the scope of HESS. The authors evidence knowledge of the topic and relevant past publications. The analyses that were undertaken are well described, as is the kNN method that is proposed. The results of the comparisons with both QR and UNEEC are clear and obviously benefit from the work that some of the authors did and published on those models. I believe that the paper is a valuable contribution to the community and recommend its publication following the clarification of some aspects discussed in the specific comments.

The authors should adopt only one version of the terms "post-processor", "postprocessor", and "post processor".

> Thank you for pointing this out. We now use the term with consistency. "post-processor" when used as an adjective and "post processor" when used as a noun.

The provided URL: www.modeluncertainty.com does not seem to work (tested on two days roughly one week apart).

> We have fixed the script and removed the bugs. The website should be able to run for files with different input variable vector sizes and for different values of k. However, the format of the files that are uploaded should be the same as that of the example files.

The chosen measure of proximity was the Euclidean distance and the different variables on which the model is conditioned upon are normalized. Whereas in a model such as QR the importance of each variable is evaluated by the model, in kNN a variable's importance is defined a priori by its scale. The possibility of attributing different ways to input variables is mentioned, but not developed nor the target of a sensitivity analysis. I have doubts that kNN performs as well as QR (for example) if the informative inputs are mixed with less informative ones (and their weights are not properly adjusted). If so, the advantages in simplicity that kNN may have risk being offset by a more demanding input selection and preparation process.

> The reviewer makes an important point about the semi-quantitative guidelines that have been employed for the choice of the input variable vector for kNN resampling i.e correlation analysis and using simple input variable vectors. Whereas, UNEEC and QR use regression to assign proper weights to their

input variables. The authors acknowledge that kNN resampling will be contingent on the choice of the variable vector, however, we also contend that the simplest of input variable vector (conditioning only on simulated system response and error in the previous time step), would be able to produce acceptable performance in capturing uncertainty. We showed this by considering such an input variable vector for three subcatchments of Severn and ten different lead times (Figure 7 and Eq 15/16). However, as the concerns are well-founded, we now explicitly state this limitation in the abstract (Page 1/ Line 22). We have added further analysis for Brue catchment using three different input variable vectors and two values of k (Page 3/ Line 31, Page 12/ Line 21-24, Figure 9 and Eq 18, 19 and 20). We also mention these results in the discussion and conclusion part to state such a sensitivity.

[Figure]

**Figure 9.  50% and** 90% prediction intervals for Brue catchment using kNN resampling. The hydrographs are shown for two different k values (99, 199) and three different input

variable vectors (Eq. (18), Eq. (19) and Eq. (20) for Input Variable Vector 1, 2 and 3 respectively). This is the largest event in the validation time series. (50% prediction interval is the interval between 25% and 75% quantiles of residual error, and 90% quantile is the interval between 5% and 95% quantiles. MPI and PICP correspond to whole validation time series.)

Line 339: Fig 5(a), the last sub-graph, the posterior distribution of the parameter ap Regarding validation metrics, it would be nice to add encompassing reliability assessment criteria such as the index α and the index ξ [e.g. Renard et al., 2010].

Thank you for this suggestion, we went through the suggested paper and have added Index Alpha to our analysis (As defined in Figure 3 of Renard et.al. 2010). We also feel that it is a valuable validation metric as it provided one number to capture the mismatch between the theoretical and observed quantiles of the error distribution. (We have added Eq 13 and Eq 14). The results are mentioned in a new Table 2.

We also went through the index ξ, and cite from the paper "Note that x = 1 does not imply perfect reliability. Consequently, this index is used primarily for detecting highly unreliable PDs." Thus, this index is useful to capture the bad pdf at the extremes. As we do not used an explicit description of a likelihood function, the authors were not sure how to evaluate the graphical description of index ξ (Figure 3 from Renard et.al. 2010). Moreover, it comes out to be perfectly 1, for a quantile resolution of 1 percent, as used in this study. The authors also feel that the information that index ξ is supposed to convey is to a great extent conveyed by Figure 6 and 10 in our manuscript. We are hopeful that the reviewer will find our line of thought adequate as a reply.

It would be important to specify in the paper whether the inputs for kNN, UNEEC and QR are the same within each catchment. If this is not the case the authors should justify how may/do differences affect results and specify what part of the comparative performances is due to model structure (kNN, UNEEC and QR) and what part can be associated with the choice of input data.

Thank you for pointing this out. As far the choice of input vector is concerned, QR values for this study uses only the predicted system response (discharge/ water level) as the covariate for the quantile. As far as UNEEC is concerned, the they used correlation analysis to choose the input variable vector. As depicted in Dogulu et. al.2016, the configuration of QR and UNEEC was chosen based on some heuristic guidelines (average mutual information and correlation analysis) and the best possible option was chosen. In this paper our comparison is restricted to such configurations of QR and UNEEC. We stay vigilant not to claim a general verdict over performance of kNN resampling compared to all configurations of QR and UNEEC. However, we do think that the analysis for three subcatchments in Severn and one in Brue substantiates the proposition that kNN performance is comparable to QR and UNEEC.

For the Brue case study, apart from two other input variable vectors (Eq. 18/19), the same input variable vector (Eq. 20) was used as in the case of UNEEC.As mentioned earlier, the results are presented in Figure 9.

                              Institute of Environmental Engineering, ETH Zürich

 The Mean Prediction Interval (MPI) results presented in Table 2 are striking. I do not believe that the discussion on why the results obtained by kNN are so strikingly better than UNEEC's or QR's justifies such a remarkable improvement and would very much like to understand the underlying reasons. Related to the previous question, I would also like to see results on the performance of the kNN method when past error observations are not added to the input vector.

We redid the analysis so that the MPI and PICP of the results can be calculated again on a less informative input variable vector, without the past errors in the input variable vector, and we got noticeable decrease in the performance of the technique. The new results are presented in Figure 9 and table 3.

We now have elaborate on the small MPI by kNN resampling in the discussion.(Page 14/Line 18-24)

Please check the References for missing information (e.g. Sikorska et al.).

Exact change made.

Technical comments

Exact change made.

**References**

[revised manuscript text omitted]

$$(6)$$

$$r_\mathrm{p} = |\boldsymbol{v}_\mathrm{p} - \boldsymbol{v}_\mathrm{t}| = \sqrt{\left[\sum_{i=1}^{n}(v_\mathrm{p}^i - v_\mathrm{t}^i)^2\right]}$$

$\boldsymbol{v}_\mathrm{p}$ is the input variable vector of the past data point in the cloud of such past data points $\boldsymbol{v}$ (Figure 1)and $r_\mathrm{k}$ is the distance to the k[th] nearest neighbour of $\boldsymbol{v}_\mathrm{t}$. Choice of the input variable vector is a problem in itself since it should include only the most relevant variables that determine the forecast uncertainty. In this study, the input variable vector is chosen based on correlation between the candidate variables and the past errors. If the correlation between the error time series and a particular candidate variable is relatively high, then it can be included in the input variable vector space. Other, more sophisticated methods involving the mutual information can be used as well (Fernando et al., 2009). This will be exemplified in the case studies described in the next section. To represent relative importance of input variables used in the search, dimensions of the input variable vector space can be suitably weighted in. Also, the model-based methods can be used where models are built for each considered candidate input variable set  and the choice is made based on their relative performance. These, however, were not explored in this study; it rather focused  on the usability of kNN search in its most basic implementation for uncertainty quantification. Nevertheless, we do demonstrate the sensitivity of the uncertainty intervals on the choice of input variable vector.

In order to level variables with different magnitudes, they are normalized. If $\sigma_\mathrm{i}$ represents the standard deviation of input variable i calculated using the past data, then:

$$(7)$$

$$r_\mathrm{p} = \sqrt{\left[\sum_{i=1}^{n}\frac{(v_\mathrm{p}^i - v_\mathrm{t}^i)^2}{\sigma_\mathrm{i}^2}\right]}$$

Once, the input variable vector space is decided, the probability of non-exceedance of a forecast error is calculated empirically by sampling from the conditional error distribution:

$$C_\mathrm{t}(e_\mathrm{t}|\boldsymbol{v} = \boldsymbol{v}_\mathrm{t}) \approx C_\mathrm{p}(e|r_\mathrm{p} \leq r_\mathrm{
[revised manuscript text omitted]

---

## Author Comment (AC2) · 23 Jun 2017

**HESSD MS No.: hess-2017-75 – Authors' Reply**

Dear Editor and Reviewers,

Please find below our revised manuscript according to the reviewers' feedback. Apart from addressing the minor comments, as a major modification, we have added new analysis to the study.

In this version one can find:

1) Additional simulation experiments for Brue catchment, where uncertainty analysis is done using three different input variable vectors and two values of k (99 and 199). This will help establish some idea of the dependence of kNN resampling performance on the choice of search space and k value.

2) Revised sentences to add proper caveats and qualifications to the performance of kNN resampling. The limitations have been stated explicitly.

We are grateful for the valuable feedback of both reviewers and the editor. We thank you for directing us towards potential improvements.

We hope that the revised manuscript meets the expectations and are thankful again for the effort of all parties involved in this review process.

Sincerely,

Omar Wani *(on behalf of the authors)*

*Please Note:* *The reference of all line numbers refers to the revised manuscript without marked changes.*

**Comment Reviewer 2 (L. Raso)**

The manuscript explores and discusses the application of k-Nearest Neighbors (kNN) method, a non-parametric machine learning technique, to estimate the predictive uncertainty in heteroschedastic streamflow forecasting. The paper is clearly written. It comes completed of a internet website where a user friendly interface makes application of kNN straightforward. The innovation is well framed in the recent literature on predictive uncertainty of heteroschedastic processes in hydrology, giving particular attention to comparable methods that estimates predictive uncertainty a posteriori. The authors clearly present advantages and limits of kNN with respect to other methods. Nonetheless, as already mentioned by the other referee, the limits of kNN in extrapolation, presently mentioned only in the conclusion, should deserve more emphasis. The manuscript brings a valid and innovative contribution to its field, and I suggest its acceptance. There are two issues, however, that could contribute to make the case for this methodology in a more convincing way, and some minor issues that deserve at least to be mentioned.

The first main issue regards the selection of the k value, i.e. the number of data points considered similar to the instance to be estimated. Fixing k is a problem of kNN method. In general, when kNN is used for prediction, k is selected in order to maximize the predictive capacity, tested by a cross-validation on data. In the manuscript the criteria for selecting k is the stabilization of residuals probability distribution. Change in residuals distribution is quantified by the cumulative difference, defined at Equation (17). The reason why the stabilization of residuals distribution is a good criteria for fixing k is not clear. Moreover, this value is monotonic, hence it does not offer a clear-cut rule. The authors propose that k is to be selected when shape changes, but this rule, differently from what stated ad page 6 line 8, is not fitted to be used in an optimisation procedure. The second issue regards the estimation of quantiles. kNN use the closest k values to build up an empirical distribution made of situations (i.e. data-points) similar to the "true" distribution that one intends to estimate. When kNN is applied for regression, the value to be predicted is the expected value, then the algorithm takes the average of k nearest data-points. In the proposed application instead, the empirical distribution is used to estimate some quantiles. Quantile estimation, however, has a different convergence rule than the expected value, particularly critical in estimating tails. Convergence rules of empirical distribution at quantiles of interest is well described in [1], chapter 21. Error in quantile estimation decreases with root square of k, and it is larger for quantiles close to 1 and 0. Using the 99th value from a set of 100 points as estimator of the 99th quantile may not be sufficient in guaranteeing sufficient convergence. Quantile estimation from empirical distribution introduce an error that must be considered, or at least discussed.

> The reviewer puts forth an important aspect of sampling errors due to the finitude of samples - the first two moments tend to less prone to errors compared to the tail estimates of a distribution. As in this study we use on 99 samples to generate quantiles corresponding to 5% and 95%, the errors are not capped by $O(1/(k)^{0.5})$. Therefore now the sensitivity of the technique to the sample size is didactically shown using two values of k (99 and 199) - its impact on PICP and MPI for Brue catchment. We have extended Figure 9. Also, we now mention this dependence on k early on in the manuscript.

                                    Institute of Environmental Engineering, ETH Zürich

(page 5/Line 24-29, page 6 line 1-2). We thank the reviewer for bringing our attention to an informative piece of literature on convergence statistics (van der Vaart, 1998). We have also added it as a reference in the manuscript.

"Moreover, this value is monotonic, hence it does not offer a clear-cut rule. The authors propose that k is to be selected when shape changes, but this rule, differently from what stated ad page 6 line 8, is not fitted to be used in an optimisation procedure."

[Figure]

**Figure 9. 50% and** 90% prediction intervals for Brue catchment using kNN resampling. The hydrographs are shown for two different k values (99, 199) and three different input variable vectors (Eq. (18), Eq. (19) and Eq. (20) for Input Variable Vector 1, 2 and 3 respectively). This is the largest event in the validation time series. (50% prediction

interval is the interval between 25% and 75% quantiles of residual error, and 90% quantile is the interval between 5% and 95% quantiles. MPI and PICP correspond to whole validation time series.)

We agree that Eq. 17 is a synthetic index and does not capture many aspects of distributional convergence. Nonetheless, the authors used it as a simple heuristic tool to agree on a reasonable value of k. The monotonicity of this index captures two aspects of the changing distribution. For small values of k, the "Cumulative Difference" changes a lot and then the sampling error decreases. However, to incorporate the concerns of the reviewer, we have removed the line referring to optimization on k Page 6, Line 16, which could have been misconstrued as an optimization exercise for k value carried out in this study. Also, as mentioned before, we have added analysis related to the sensitivity of k (Figure 9). We don't seem to notice worrying changes in the PICP and MPI of the Brue catchment when changing k from 99 to 199. However, as expected, the MPI does get somewhat bigger. We discuss this dependence explicitly on Page 14/Line 20-24.

In the discussion on verification index, the authors show that they are aware of thelimits in using few indicators. The authors state that "PICP and MPI [...] give a reasonable assessment of performance". But this is not further explained. There are likely good reasons to select these indicators, but this should be better explained in the text, considering also that the application is about flood forecasting.

The idea of checking PICP at 90 uncertainty bands is a common practice in hydrology. However, more tail events might become interesting for design problems, then the reliability of the whole error distribution is more interesting than the mere computation of PICP and MPI. Taking the advice of reviewer 1, we have also added another metric for performance – the Index Alpha (Eq. 13/14). And the results are presented in Table 2.

In Equation 7, variables are standardized one at a time, losing information about covariance. Why not considering variables as a multidimensional distribution, then using the covariance matrix to standardise? This would make use of the mutual information about variables in a more efficient way.

The individual normalization prevents the kNN conditioning to be exclusive to the dimension with higher variance. However, as the reviewer points out, there can be more advanced ways of normalization, like the usage of covariance matric which captures the linear dependence between different dimension of the input variable vector. We have not used tested techniques in this research. We are aware that better metrics to choose of input variable vector and for the normalization can improve the technique significantly. Hopefully in the future we can carry out more analysis to better delineate such technique.

**Other comments**
Page 3, line 24: add "than" after simpler

Exact change made.

Page 5, line 12: "uncertainty in observational data is not considered", why can not it be included?

> Exact change made.

Page 8 line 23: remove extra dot.

> Exact change made.

Page 9 line 17: the adverb "just" looks like non necessary.
> Exact change made.

Page 10 line 11: The result description would be easier to follow if the reference to the figure was placed at the beginning of this paragraph (from line 19 to line 11).
> Exact change made.

**References**

[revised manuscript text omitted]

$$r_\mathrm{p} = |\boldsymbol{v}_\mathrm{p} - \boldsymbol{v}_\mathrm{t}| = \sqrt{\left[\sum_{i=1}^{n}(v_\mathrm{p}^i - v_\mathrm{t}^i)^2\right]} \tag{6}$$

$\boldsymbol{v}_\mathrm{p}$ is the input variable vector of the past data point in the cloud of such past data points $\boldsymbol{v}$ (Figure 1)and $r_\mathrm{k}$ is the distance to the k$^\mathrm{th}$ nearest neighbour of $\boldsymbol{v}_\mathrm{t}$. Choice of the input variable vector is a problem in itself since it should include only the most relevant variables that determine the forecast uncertainty. In this study, the input variable vector is chosen based on correlation between the candidate variables and the past errors. If the correlation between the error time series and a particular candidate variable is relatively high, then it can be included in the input variable vector space. Other, more sophisticated methods involving the mutual information can be used as well (Fernando et al., 2009). This will be exemplified in the case studies described in the next section. To represent relative importance of input variables used in the search, dimensions of the input variable vector space can be suitably weighted in. Also, the model-based methods can be used where models are built for each considered candidate input variable set  and the choice is made based on their relative performance. These, however, were not explored in this study; it rather focused  on the usability of kNN search in its most basic implementation for uncertainty quantification. Nevertheless, we do demonstrate the sensitivity of the uncertainty intervals on the choice of input variable vector.

In order to level variables with different magnitudes, they are normalized. If $\sigma_\mathrm{i}$ represents the standard deviation of input variable i calculated using the past data, then:

$$r_\mathrm{p} = \sqrt{\left[\sum_{i=1}^{n}\frac{(v_\mathrm{p}^i - v_\mathrm{t}^i)^2}{\sigma_\mathrm{i}^2}\right]} \tag{7}$$

Once, the input variable vector space is decided, the probability of non-exceedance of a forecast error is calculated empirically by sampling from the conditional error distribution:

$$C_\mathrm{t}(e_\mathrm{t}|\boldsymbol{v} = \boldsymbol{v}_\mathrm{t}) \approx C_\mathrm{p}(e|r_\mathrm{p} \leq r_\mathrm{k}) = j/k \tag{8}$$

[revised manuscript text omitted]

---

## Editor Comment (EC1) · B. Schaefli (Editor) · 26 Jun 2017

Both reviewers state that this is a good quality manuscript of interest to the readership of HESS. They both also gave some detailed comments on how to further improve the manuscript. The authors answered all of the raised points and I would like to invite them to resubmit a revised version according to their detailed responses. The revised version will not go to a further review round (editor review only).

---

## Author Comment (AC3) · 27 Jun 2017

[revised manuscript text omitted]
_{\rm t}(e|\boldsymbol{v}=\boldsymbol{v}_{\rm t}) = C_{\rm p}(e|\boldsymbol{v}=\boldsymbol{v}_{\rm t}) \tag{4}$$

The subscript p denotes historical time series. Therefore  $C_p$  is the cumulative distribution function of residual errors from the 15 past. In Eq. (4),  $C_p$  is being conditioned to the input variable vector at time t. Nevertheless, as we only have single realizations of the error variable *E* for each historical point, we relax the constraint of  $\boldsymbol{v} = \boldsymbol{v}_t$ . Instead, we assume that the nearby neighbours of  $\boldsymbol{v}_t$  in n-dimensional space will have a similar probability distribution of errors as  $\boldsymbol{v}_t$  and that these historical errors are samples from  $C_p(e|\boldsymbol{v} = \boldsymbol{v}_t)$ . An empirical probability distribution can thus be constructed using the kNN historical errors:

$$C_{\rm t}(e_{\rm t}|\boldsymbol{\nu}=\boldsymbol{\nu}_{\rm t}) \approx C_{\rm p}(e|\boldsymbol{r}_{\rm p} \leq \boldsymbol{r}_{\rm k}) \tag{5}$$

20 where  $r_p$  is the Euclidean distance in n-dimensional space of input variables.

$$\boldsymbol{r}_{\rm p} = |\boldsymbol{v}_{\rm p} - \boldsymbol{v}_{\rm t}| = \sqrt{\left[\sum_{\rm i=1}^{\rm n} (v_{\rm p}^{\rm i} - v_{\rm t}^{\rm i})^2\right]}$$
(6)

 $v_p$  is the input variable vector of the past data point in the cloud of such past data points v (Figure 1)and  $r_k$  is the distance to the kth nearest neighbour of  $v_t$ . Choice of the input variable vector is a problem in itself since it should include only the most

relevant variables that determine the forecast uncertainty. In this study, the input variable vector is chosen based on correlation between the candidate variables and the past errors. If the correlation between the error time series and a particular candidate variable is relatively high, then it can be included in the input variable vector space. Other, more sophisticated methods involving the mutual information can be used as well (Fernando et al., 2009). This will be exemplified

- 5 in the case studies described in the next section. To represent relative importance of input variables used in the search, dimensions of the input variable vector space can be suitably weighted in. Also, the model-based methods can be used where models are built for each considered candidate input variable set and the choice is made based on their relative performance. These, however, were not explored in this study; it rather focused on the usability of kNN search in its most basic implementation for uncertainty quantification. Nevertheless, we do demonstrate the sensitivity of the uncertainty intervals on
- 10 the choice of input variable vector.

20

In order to level variables with different magnitudes, they are normalized. If  $\sigma_i$  represents the standard deviation of input variable i calculated using the past data, then:

$$\boldsymbol{r}_{\rm p} = \sqrt{\left[\sum_{\rm i=1}^{\rm n} \frac{(v_{\rm p}^{\rm i} - v_{\rm t}^{\rm i})^2}{\sigma_{\rm i}^2}\right]} \tag{7}$$

Once, the input variable vector space is decided, the probability of non-exceedance of a forecast error is calculated 15 empirically by sampling from the conditional error distribution:

$$C_{\rm t}(e_{\rm t}|\boldsymbol{\nu}=\boldsymbol{\nu}_{\rm t}) \approx C_{\rm p}(e|\boldsymbol{r}_{\rm p} \le \boldsymbol{r}_{\rm k}) = j/k \tag{8}$$

[revised manuscript text omitted]

(EA, 2013 and Marsh, T. J. and Hannaford, J., 2008)

Table 1: Basin information for Upper Severn subcatchments

Table 2: Index Alpha (α) for different lead times of Upper Severn subcatchments

| Lead Time (h) | 1    | 12   | 24   | 48   |   |
|---------------|------|------|------|------|---|
| Llanerfyl     | 0.92 | 0.87 | 0.79 | 0.64 | - |
| Llanyblodwel  | 0.93 | 0.95 | 0.93 | 0.90 |   |
| Yeaton        | 0.97 | 0.94 | 0.94 | 0.75 |   |

|             | PICF  | PICP (Expected 90%) |       | MPI $(m^3/s)$ |      |      |
|-------------|-------|---------------------|-------|---------------|------|------|
|             | UNEEC | QR                  | kNN   | UNEEC         | QR   | kNN  |
| Calibration | 91.19 | 90.00               | 86.3  | 1.58          | 1.69 | 0.51 |
| Validation  | 88.29 | 82.33               | 84.42 | 1.37          | 1.39 | 0.21 |

Table 3: Performance of various uncertainty estimation techniques for Brue catchment. For kNN resampling andUNEEC the same input variable vector is used (Eq. (20)). For QR only  $Q_{sim}$  is used.

Figure 1. Dependence of error samples on the value of k. For larger values of k, points are at a greater distance from  $v_t$  (the prediction step), thus compromising the conditioning of the residual error probability distribution on  $v_t$  (Eq. 15).

Figure 2. Upper Severn subcatchments with gauging stations (From López López et al., 2014)

5